# The Splendour of Glitter: Silver Leaf in *barniz de Pasto* Objects

**María Cecilia Álvarez-White [1]**, **David Cohen [2],*** and **Mario Omar Fernández [2]**

1   Conservator in Private Practice, Bogotá 110221, Colombia; mca4@hotmail.com
2   Laboratorio de Estudios de Artes y Patrimonio—LEAP, Universidad de los Andes, Bogotá 111711, Colombia; mo.fernandez@uniandes.edu.co
*   Correspondence: d.cohen@uniandes.edu.co

**Abstract:** This paper presents current findings about the historical use of silver and metallic elements as part of the decorative technique from Colombia known as *barniz de Pasto*. The results are part of broader ongoing research. The aim is to interpret and assess the introduction of metal as a significant component in the consolidation of the technique. Our research uses well-established imaging techniques (visible light macrophotography; infrared and ultraviolet photography), cross-sections, and elemental mapping from SEM-EDS. The results are compared and interpreted within a holistic framework suggesting the introduction of silver as part of an innovation of the *barniz de Pasto* technique, either as whole silver leaf sandwiched between layers of mopa-mopa or as "venturine", obtained by kneading, and therefore fragmenting, silver leaf within mopa-mopa layers.

**Keywords:** *mopa-mopa*; *barniz de Pasto*; silver leaf; conservation sciences; venturine

## 1. Introduction

The pre-Hispanic communities of the department of Nariño in Colombia produced beaded necklaces made with a natural resin, which was also known as *mopa-mopa* [1]. It comes from the plant *Elaeagia pastoensis* L.E Mora, which belongs to the Rubiaceae family and grows wild in the south-western rainforest of the country. During the early colonial period and later, (with the consolidation of the viceroyalty of New Granada at the start of the 18th century), the technique evolved to include new materials, forms, and designs in response to high demand for these artifacts. The production of the wooden pieces decorated with *barniz de Pasto* further expanded during the 19th and 20th centuries, and artisan *maestros* in the city of Pasto even today continue the tradition. This long trajectory in the transmission and innovation of knowledge and skills was recognised by UNESCO as Intangible Cultural Heritage of Humanity in December 2020 requiring safeguarding.

However, despite the cultural and artistic importance of these objects, few technical studies have been carried out to characterise materials [2–5] and identify their transformation over time. In particular, the lack of research with respect to the introduction and use of materials such as silver leaf and brass is evident.

Added to the above shortcomings is the absence of a historical documentary foundation, as nowadays only general descriptions made by chroniclers from the colonial period and travellers of the 19th century are available, with few exceptions [6]; there are also no known dates or inscriptions that would allow a direct dating of the objects.

Another circumstance which has an impact on the knowledge and dissemination of the technique is that most of the extant works belong to private collections in both Colombia and abroad and only in few cases are they accessible to the public. Moreover, museums usually lack the range of representative materials that would allow comparative technical studies to be carried out.

The current research is part of an ongoing wider project, led by the conservator María Cecilia Álvarez-White with the support of the Laboratorio de Estudios de Artes

y Patrimonio (LEAP), of the Faculty of Arts and Humanities and the Vice-presidency of Research and Knowledge Creation of the Universidad de los Andes.

The aim is to show, in an exploratory manner, the use of metal leaf in the development of the *barniz de Pasto* technique based on the evidence from the laboratory analyses, which will provide a framework for interpretation that will support its further in-depth study.

The effect of glitter on the objects decorated with this technique may have emerged, in our own opinion, from the mid-17th century onwards. Its greatest heyday, however, dates from the 18th century and, with some variations, extends throughout the 19th and 20th centuries to the present day.

The first metallic glitter was achieved thanks to the combined handling of both the metal leaf and the *mopa-mopa*, which is gradually applied in an extensive way until the ornamented surfaces are covered in an iridescent shimmer. Later, in the 19th century, brass leaf, consisting of an alloy of copper and zinc, was introduced to these effects to add a copper tone to the decoration. A Spanish translation of this paper is available in Supplementary Materials.

## 2. Materials and Methods

For the collection and processing of data, a protocol for approaching the pieces was proposed in which the available objects were numbered with a unique code which identified them. In the first instance, they were classified into groups that showed common visual characteristics on the basis of the following variables:

1. The thickness and relief of the *barniz* decorations;
2. The opacity or shine of the surface;
3. The appearance of vibrant colours;
4. The presence of metallic glittery elements.

A total of 230 objects decorated with *mopa-mopa* were examined, of which 91 were analysed with different techniques in order to explore aspects of their materiality and to characterise the metallic elements that give the decorations their shine. Subsequently, representative objects were chosen from each group for the final selection of 41 objects.

A detailed photographic record of these artworks was made using visible light, ultraviolet light-induced fluorescence (UVF), and infrared (IR) photography. For the visible and ultraviolet photography and the fluorescence, a Canon® EOS 7D Mark II camera (Canon, Tokyo, Japan) was used with a SPECTROLINE® Tritan 365 UV lamp (Spectroline, New York, NY, USA) and a UV TIFFEN filter® (TIFFEN, Oklahoma City, OK, USA). The IR photographs were taken using a Cultural Heritage Science Open Source—CHSOS (CHSOS, Viagrande, Italy) modified NIKON® D7100 camera (NIKON, Tokyo, Japan). An ELINCHROM® D-Lite LX-4 lamp (ELINCHROM, New York, NY, USA) was used as the light source.

These methods allowed the identification of some of the metallic decorations, especially in areas where the deterioration or restoration of the objects made it difficult to view them.

Based on this first approach and the analysis of the information, the objects with metallic shine were classified to finally select the 11 objects on which to carry out the laboratory analyses. From these, 2 or 4 samples were taken to prepare cross-sections and for elemental analysis. The samples were embedded in acrylic resin for handling and were morphologically analysed with an optical microscope. They were then taken to the TESCAN FE-MEB LYRA3 scanning electron and focused ion beam microscope of the MicroCore Laboratory. The SEM has an integrated X-ray energy dispersive spectroscopy microanalysis system (EDS) for elemental identification and mapping. The quantitative analysis has an accuracy of $\pm 2\%$, with detection limits of approximately 100 ppm.

The accounts of chroniclers and travellers, the only reference documents known to date, were consulted, to provide insight about the manufacturing sites of the objects and the description of some of their details. Of particular note are the accounts of Fray Pedro Simón (1574–1628), who was perhaps the first to mention the resin-producing plant, as well

as the objects that were decorated with it and the places of production, pointing to Mocoa, Quito, and even "other parts of Peru" where the Inka wooden ceremonial vessels (known as "Qeros") decorated with *mopa-mopa* were made [7] (p. 273).

Moreover, other chroniclers who provide complementary information were consulted, such as Lucas Fernández de Piedrahita (1624–1688) [8], Jorge Juan and Antonio de Ulloa (1749) [9], Miguel de Santisteban (1691–1776) [10], Juan Magnin, S. J. (1701–1753) [11], Fray Juan de Santa Gertrudis (1724–1799) [12], and Juan de Velasco (1727–1792) [13].

In the early 19th century, more concrete data on the materials and ways of working can be found in the detailed descriptions made by travellers such as Alexander von Humboldt (1769–1859), who in 1801 visited the city of Pasto and recorded in his diaries interesting aspects about the materials used and the manufacturing technique, describing, among other things, details such as: "*the shadows are obtained by placing one membrane on top of the other*" [14] (p. 127a).

The botanist José Celestino Mutis (1733–1808), who from 1783 oversaw the Botanical Expedition of the New Kingdom of Granada, mentions several pigments such as orpiment as well as natural dyes used in the *barniz de Pasto* decoration [15].

Colonel John Potter Hamilton (1777–1873) arrived in the country in 1823, a few years after the declaration of independence, and praised the *totumas* (gourds) adorned with brightly coloured flowers [16]. The French chemist Jean Baptiste Boussingault (1801–1887) describes in his memoirs "wooden objects such as gourds, boxes and containers for storing wine or brandy" decorated with the resin and details its application technique and some of its chemical characteristics [17] (p. 460).

Edouard André (1840–1911), when referring to resin, emphasises its "extraordinary consistency, it resists cold, heat and water and adheres strongly to wood" [18] (p. 755) and includes an engraving showing a workshop of that time.

The North American botanist and tropical flora scholar, Isaac Farewell Holton (1812–1874) describes *totumas* and other objects made in Pasto with the *mopa-mopa* "brought from the distant headwaters of the Amazon" [19] (p. 538).

## 3. Results

### 3.1. The Barniz de Pasto Technique before the Introduction of Metals

Before the arrival of the Spaniards, the indigenous people of the Pasto area knew about the resin, the place where the plant that produces it grew, the times when the buds were harvested and how they were processed. They produced necklace beads from *mopa-mopa* which had symbolic and ceremonial use. Their documentation is associated with burial contexts carried out in archaeological excavations at the San Agustín Park, in the current department of Huila [20] and in the village of Miraflores, in Nariño [21].

In the beads, not only was the resin's natural colour used, which presently has a brown appearance, perhaps due to natural ageing because it is an organic material (Figure 1), but also two other shades: one with a light finish due to the use of calcium carbonate, possibly lime (SEM analysis carried out by Fernández, M.O.), and another with a dark shade, achieved with a dye of organic origin, which has yet to be identified.

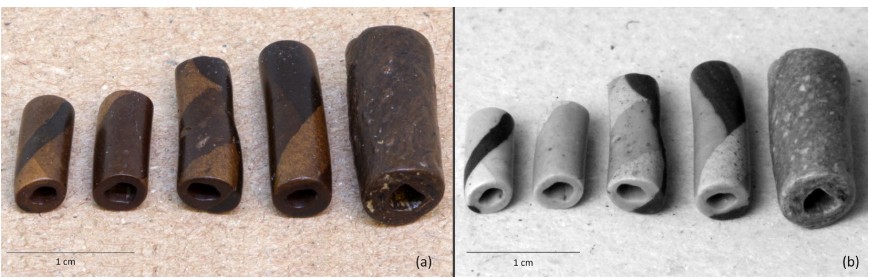

**Figure 1.** Group of necklace beads with normal (**a**) and infrared (**b**) lighting. The four on the left show interspersed resin bands, two of them coloured. The largest bead, on the far right, is made from the resin in its natural, undyed state. Private collection. (Photographs: Cohen, D).

The bands are produced with thin sheets of resin, which, while still hot, were stretched and wound diagonally around a circular wooded core. They were then cut to the desired size, up to 3 centimetres long. When they cooled, they acquired their final consistency. (Figure 1). The sheets have a thickness of approximately 2 to 3 millimetres.

The pre-Hispanic necklace beads, the only items from that period made from *mopa-mopa* resin known to date, represent the precursor of the decorative technique that is known today as *barniz de Pasto*.

### 3.2. The Start of the Colonial Period

During the early colonial period (possibly towards the end of the 16th and the beginning of the 17th century) when the Spaniards arrived, *mopa-mopa* began to be used to decorate wooden objects produced based on European models. The demand for these pieces generated a market for works that constituted elements of prestige for the new social class linked to political and ecclesiastical power. They were made of cedar wood, abundant in the area of Pasto, just as described in the 18th century by the friar Juan de Santa Gertrudis (vol. 3, p.110). The analysis of the wooden support of a small table cabinet owned by the Victoria and Albert Museum (carried out by Dr. Peter Gasson of the Royal Botanic Gardens Kew) in London corroborates the chronicler's statement [22].

Pieces made in what has been called the early period for this research are characterised by a limited chromatic range with a matte appearance, although the objects retain the satin shine of the resin. The decoration displays elements from the indigenous tradition intermingled with vegetation and animals from the surrounding environment, as well as reinterpretations of European figures themselves, such as the Habsburg double-headed eagle.

The wood is completely covered with a transparent resin film which has a protective function and on which the decoration is applied. The sheets are tinted with dyes of organic origin and with powdered *albayalde* (the pigment lead white) when light shades are desired. In these works, the resin sheets, much thinner than the ones used for the necklace beads, have a thickness of approximately 50 µ to 200 µ. The pre-cut membranes are applied to the generally dark, cherry red or olive green-coloured background. They are stretched one after the other to fill in the figures, or simply applied as contour lines or even loosely, as scrolls. (Figure 2).

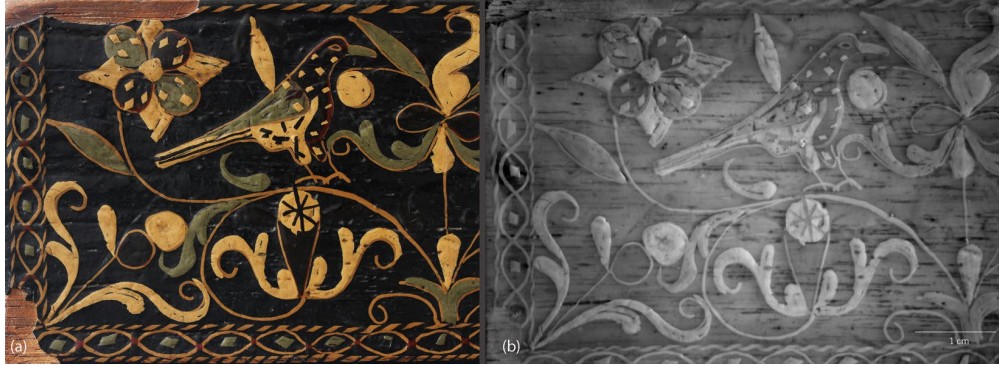

**Figure 2.** The light areas, resulting from mixing the resin with *albayalde*, have an opaque finish. The cherry red-coloured areas, such as the bird's chest, were produced with a transparent membrane dyed with an organic dye as seen in figure (**a**). On the right (**b**), under infrared light, we can see the slight relief of the applications that make up the designs, as well as the small resin cut-outs with which the flower petals were filled in the upper left corner. (Photographs: Cohen, D). Private collection.

The calcium carbonate used in the pre-Hispanic period is now substituted by the *albayalde*, which arrives through trade from the port of Sevilla [23] and provides an opaque appearance when mixed with the resin. This pigment was skilfully handled to achieve gradations of light shades when used in different concentrations (Figure 3). Lead white was characterized as part of the decoration of Peruvian Inka "Qeros" as well [24]. It is possible

that, in these instances, a vegetable dye, such as the root of the *escobedia* or "saffron of the earth", as mentioned by Humboldt (p. 127a), may have been added in order to achieve the ochre-like yellow colour.

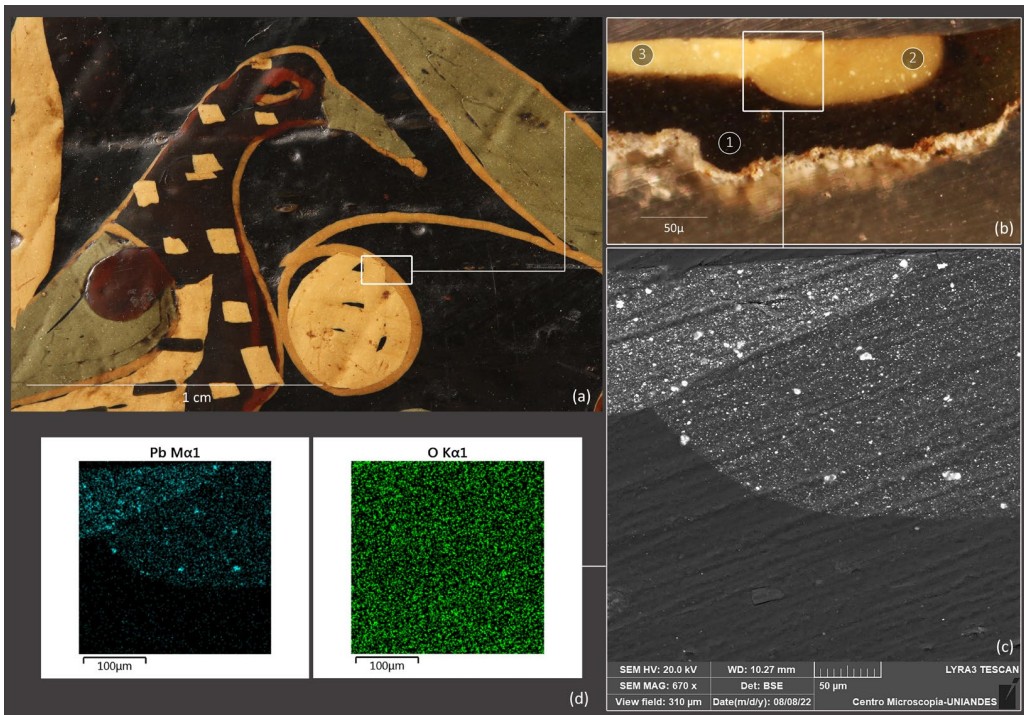

**Figure 3.** (**a**) Detail of the decoration made with *mopa-mopa* sheets on the dark-coloured base that serves as a background for the composition. (Photograph: Álvarez-White, M.C). In layers 2 and 3 of the stratigraphic section (**b**) two different concentrations of the white pigment (*albayalde*) can be observed. Layer 1 corresponds to the resin which is the foundation of the design. In the SEM image (**c**) the white dots of layers 2 and 3 confirm the presence of lead white in the elemental mapping (**d**). (Analysis and photography: Fernández, M.O). Private collection.

The dark shade is achieved with calcined or highly concentrated indigo, which sometimes also serves as a general background for the designs. The greens are obtained by mixing the blue of the indigo with the extract of a plant, not yet identified, rich in yellow flavonoid-based components with high levels of quercetin and kaempferol [25]. The red material, still to be characterised by chemical analysis, could be cochineal [26], given its appearance. It is necessary to bear in mind that as the pieces have undergone natural processes of ageing, the visual perception of the colours has also changed.

### 3.3. Introduction of Silver Leaf

As the colonial period progressed, objects with more complex ornamentation began to be made, as the use of pigments such as lead white was enriched with the introduction of both silver leaf and a white pigment with an opaque finish known as calomel. This mix of materials creates a decoration that interweaves shiny elements with matte ones.

In these later pieces, more care is taken in the processing of the resin sheets, which now become even thinner, reaching a thickness of approximately 30 µ to 50 µ. The chromatic range is broadened with the introduction of the metal: a clear iridescent shade is obtained when the membrane that covers it is used in its natural shade; or by adding other shades of different colours to it, when it is dyed with organic dyes. This period saw the first appearance of backgrounds complete with metallic glitter, onto which the designs are applied (Figure 4).

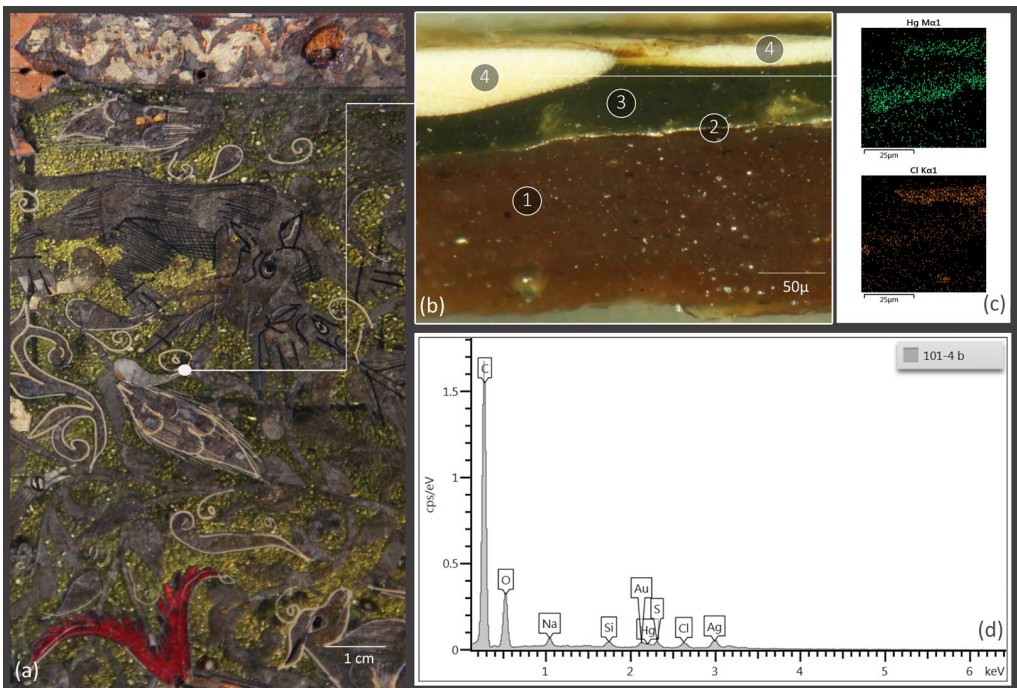

**Figure 4.** Stratigraphic section at 200× (**b**) taken from a white strip of the ornamentation (**a**). The white dot indicates the sampling location (Photograph: Álvarez-White, M.C). In image (**b**), Layer 4 shows the presence of mercury (Hg) and chlorine (Cl), which confirms the calomel pigment. The silver leaf is visible in layer 2, while layer 1 is the base resin and layer 3 the green coloured resin. (**b**). The EDS spectrum (**d**) shows mercury and chlorine detected in the elemental mapping (**c**). The presence of gold (Au) is due to the coating process of the sample for SEM analysis. (Analysis and photography: Fernández, M.O). Private collection.

The borders framing the compositions are often made of tinted silver leaf and within them are Nanban-influenced designs [27], such as those found on the casket of the Victoria and Albert Museum in London (W.7–2018). Recent analysis of this piece has revealed the presence of calomel used in the fine white lines with a matte finish, according to Humphrey et al. (p. 150).

Calomel ($Hg_2Cl_2$) or mercury white is a highly toxic synthesised pigment that has not been found in South America in art objects, with the exception of objects made with the *barniz de Pasto* technique. During several research projects, such as the ones already mentioned by Pozzi et al. [3], Sánchez L. [4], Burgio et al. [5], and Álvarez-White [6], its use in the decoration of different pieces has been confirmed also by Romero [28].

During the colonial period, quicksilver, as mercury was also known, was primarily used for processing of gold and silver. Barba mentions its use as early as 1574 in the mines of Potosí in Bolivia [29] (p. 71). In 1789, Juan de Velasco mentions how in the mountains of Cuenca, Ecuador, there was a quicksilver mine whose production supplied other provinces, but it had to be closed because of the danger it posed to the "health of the Indians" (vol. 1, p. 45). More recent works show that it was used in Europe during the 15th and 16th centuries to illuminate manuscripts and miniatures [30].

In most of the pieces that feature white designs made with calomel, other layers with an opaque appearance can be observed, as well as decorations made with silver leaf covered by membranes of *mopa-mopa*. These may or may not be coloured and may display a golden, green, or blue shade in order to enhance the iridescence of the metal underneath. (Figure 5).

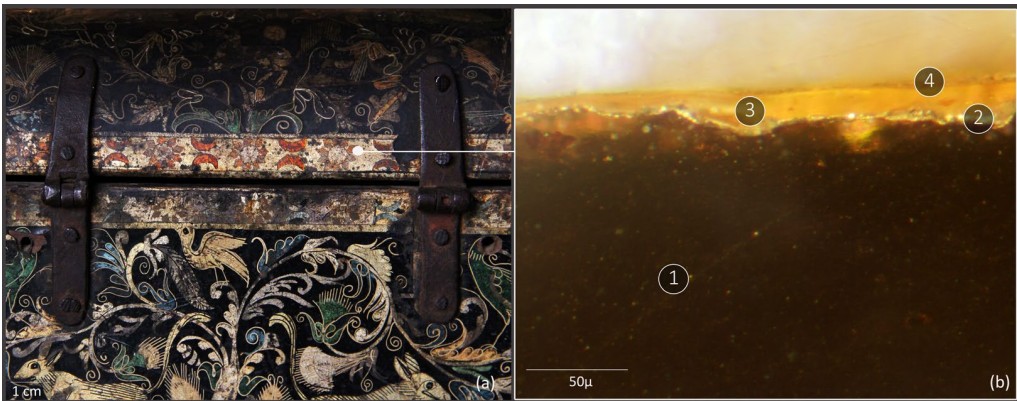

**Figure 5.** Some elements made in various iridescent colours, through the use of silver leaf (**a**), are an example of the introduction of designs with a metallic finish mixed with others with a matte appearance. Calomel, which has an opaque finish, is applied either in the form of thin white strips to delimit the contours or in the form of lines. The white dot in image (**a**) indicates the sampling location (Photograph: Álvarez-White, M.C). The stratigraphic section (400×) (**b**) shows the resin layer serving as background (1), then the silver leaf (2), and on top of the latter, a 15 μ thick layer of resin (3). Over the top, there is a thin layer of accumulated dirt (4) (Analysis and photography: Fernández, M.O). Private collection.

As part of the *barniz de Pasto* technique, the silver leaf is handled as a complex structure called a "composite membrane", according to Álvarez-White (2023, p. 168). It involves extending the metallic leaf on the first sheet of *mopa-mopa* and then covering it with another *mopa-mopa* layer that can be tinted with an organic powder dye. In this way, the metal preserves its metallic lustre while avoiding corrosion processes, which only occur when the metal is exposed through wear or use, therefore blackening due to oxidation.

These composite films were spread on a wooden surface, previously covered with a background layer of resin. In 1771, Father Cicala described this procedure as follows: "when the craftsman wants a certain field or flower to look like silver, after painting the first layer [resin], he spreads out the silver leaf and on top of this he paints another layer of varnish in the colour he wants" [31] (p. 514). This decorative technique is still employed today in a similar fashion.

It should be noted that, since the pre-Hispanic period, processing the resin involved chewing it, as mentioned by Fray Juan de Santa Gertrudis (vol. 3, p. 110) and Humboldt (p. 127a), who points out that saliva played an important role in achieving homogeneous sheets and better amalgamation with the dyes and pigments such as lead white. Even in the mid-twentieth century, Alfonso Zambrano, a renowned master woodcarver in the city of Pasto, mentioned how, around 1940, the varnishers who came to his workshop to commission small wooden furniture (Figures 6 and 7) "had very worn teeth and could be recognised from a distance by the muscular and powerful appearance of their jaws" [32] (p. 21).

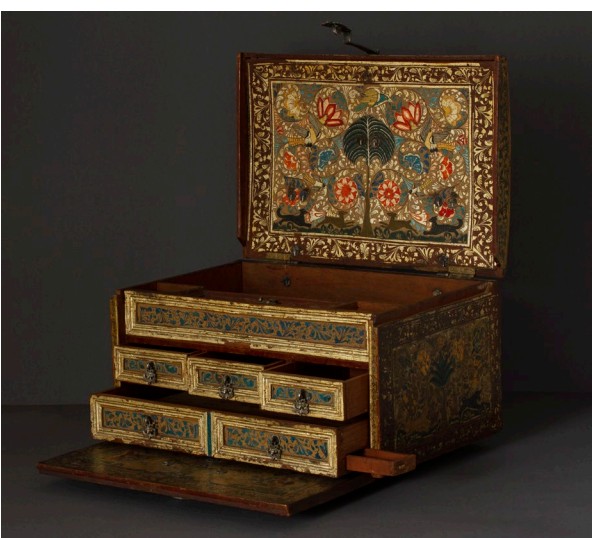

**Figure 6.** Example of the configuration of a typical *escritorio*, common from the colonial period to the 18th century. Objects of this type are made of wood and equipped with drawers where there are often secret compartments. All exterior surfaces are also ornamented with *barniz de Pasto.* The lid and fall front are foldable. "Escritorillo de la palmera", Open dimensions: 46 × 39.5 × 26 cm. Private collection. (Photograph: Álvarez-White, M.C).

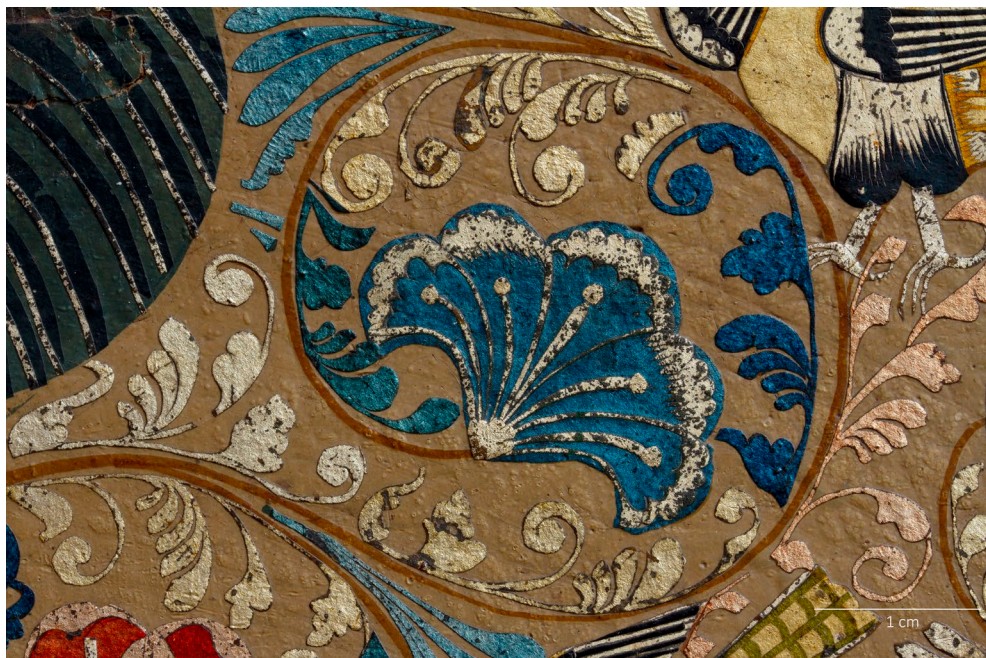

**Figure 7.** This raking light photograph allows us to appreciate the natural relief caused by the superimposition of layers on the matte background, which is made with a mixture of *mopa-mopa* resin and lead white. In the composite membranes, the colours used to dye the resin covering the silver leaf are skillfully balanced to achieve the iridescent effect. "Escritorillo de la palmera". Private collection. (Photograph: Álvarez-White, M.C).

### 3.4. Iridescent Effects

According to the descriptions by chroniclers and travellers since the 18th century, objects decorated with *barniz de Pasto* are uniquely characterised by their lustre, which is compared favourably to that of Chinese porcelain (Fray Juan de Santa Gertrudis 1970, vol. 1, p. 175).

Subsequently, designs using silver leaf included opaque backgrounds made with resin mixed with lead white. The silver leaf increased the object's luminosity and allowed for further diversification of the colour palette. The iridescent white of the silver metal complemented the colouring.

Although not all the dyes used have been identified yet, indigo is still used in various concentrations. It is possible that it is the same indigo (*Indigofera suffruticosa*—which is known as the American species) that, according to Humboldt (p. 127a), was used calcined or highly concentrated to obtain a black colour. Bruquetas [33] (p. 163) explains that this dye was shipped from America to Spain in large quantities. It is probable that the mixture of resin with *escobedia*, due to its yellow colour, is what gives the golden finish to the silver metal. The use of other organic dyes such as cochineal and achiote (*bixa orellana*) has not been ruled out. (Figure 8).

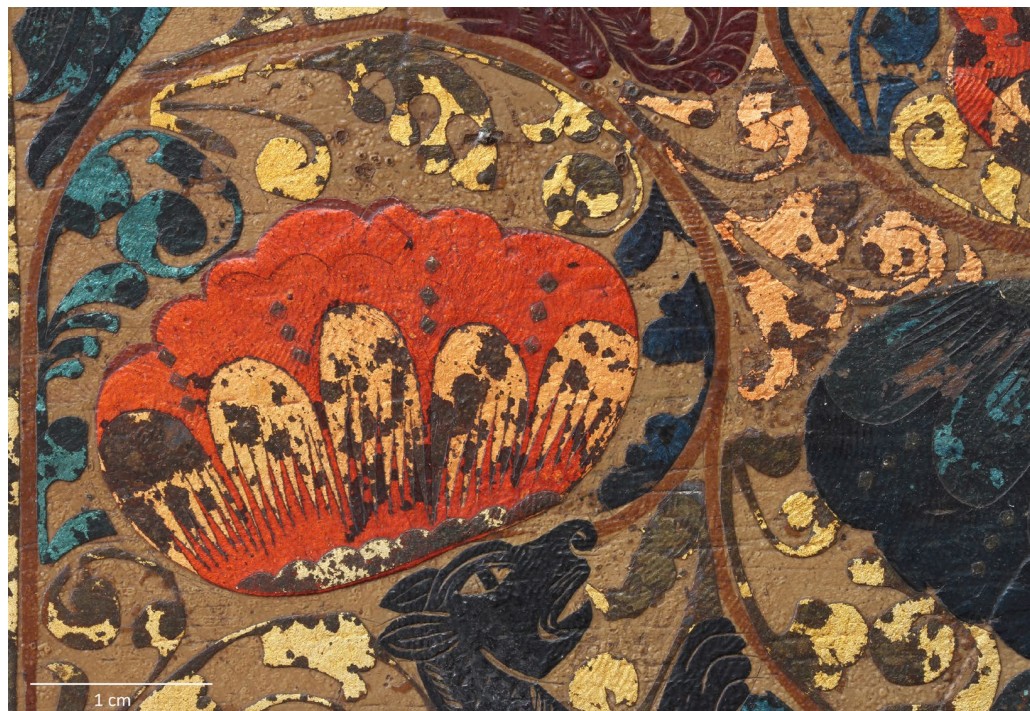

**Figure 8.** Detail of the back cover of a small table cabinet with metallic decorations in different tones, including gold, a result of the effect produced by the yellow-coloured resin membrane. Fingerprints can also be observed. The overall background is a matte layer of resin (Photograph: Álvarez-White, M.C.). Private collection.

White made from mercury is no longer used as part of the technique. It is unclear when exactly its use ended but white made from lead, with its matte appearance, took its place as the background of the compositions.

It is common to find fingerprints left by the manual pressure from fixing the membranes while they were still hot (Figures 8 and 9). The thinness of these films, which are approximately 10 μ to 30 μ thick, is what possibly led Miguel de Santisteban in 1740 to refer to them as "onion peel" [34] (p. 125) and Juan de Velasco in 1789 to describe them as "thinner than the thinnest paper in China" (vol. 1, p. 56) (Figure 10).

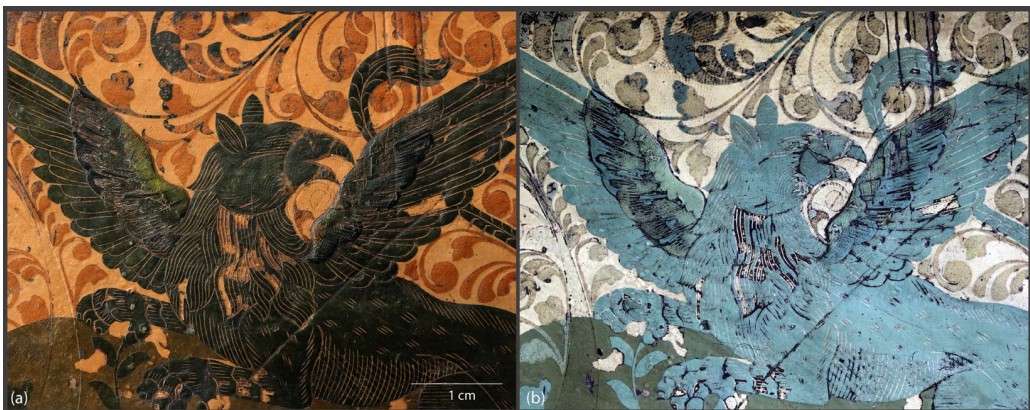

**Figure 9.** Detail of a mythical figure—a griffin—applied on a matte background (**a**) (Photograph: Álvarez-White M.C). Under UV light (**b**), the material that gives the figure of the griffin its green colour acquires a turquoise colour. The vegetal motifs—scrolls—in the background, a good part of which have lost their tinted silver leaf coating but were originally tinted dark blue, can be seen in a light grey tone. Remains of coloured metal stand out in dark tones that enhance details in the wings of the animal and in some scrolls. The general background of the composition, produced by mixing resin with lead white, fluoresces white. (UVF photograph: Cohen D.). Private collection.

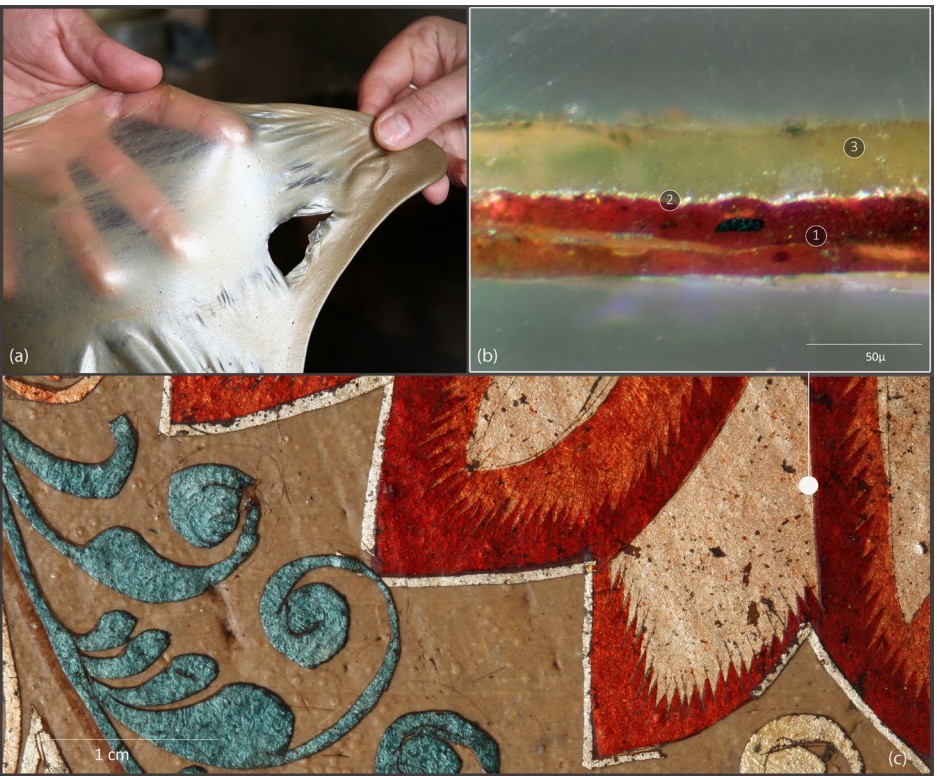

**Figure 10.** The delicate membranes of resin described in the 18th century are made in a similar way today (**a**). In the stratigraphic section (400×) (**b**), the yellow-tinted resin (3) layered over silver leaf (2) is visible. Layer 1 corresponds to the red-coloured resin decoration. The subtly tinted red resin membranes overlap to achieve the shading that Humboldt describes (**c**). The white dot in the image (**a**) indicates the location where the sample was taken. (Photographs (**a**,**c**): Álvarez-White, M.C) (Analysis and photograph: Fernández M.O). Private collection.

Another way of using the silver leaf, as far as we know exclusively used as part of the *barniz de Pasto* technique, was mentioned by Humboldt in 1801 as 'venturine' (p. 127a). It is likely that Humboldt named the technique this way because of its resemblance to the

optical effect presented by the semiprecious stone aventurine, a variety of quartz, whose mica inclusions cause an iridescent effect similar to that of silver leaf. The 'venturine' is obtained by using the silver leaf between two *mopa-mopa* membranes and, unlike the traditional way of use, kneading it after adding another bit of resin. As a result, the metal is ground within the mass and is observed as small, scattered metallic fragments. The mixture is then stretched to form a film on the wooden surface on top of which designs of the desired composition are applied. This is how the background acquires a particular sparkling finish (Figures 11–13) whose appearance is different from that exhibited by the silver leaf when it is used as a sheet sandwiched between the *mopa-mopa* membranes.

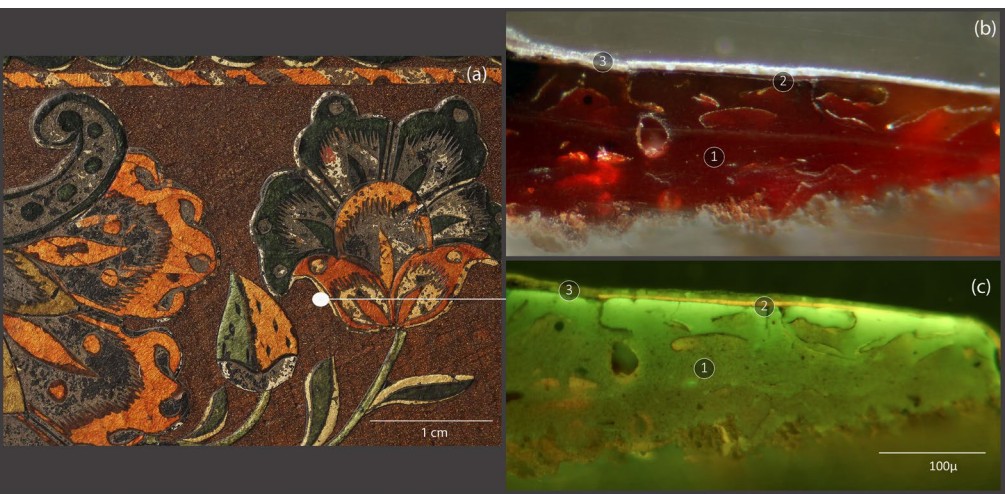

**Figure 11.** (**a**) In this figure, one can observe the sparkling brightness of the reddish-brown background achieved by the 'venturine', on top of which the designs were applied. The white dot indicates the sampling location (Photograph: Álvarez-White, M.C). The stratigraphic section (**b**) shows the red-coloured, lower resin background with silver particles corresponding to the technique named by Humboldt as 'venturine' (1), over which other layers (2 and 3) were applied. Layer 3 corresponds to a coloured silver leaf, as explained before (Figure 5). Image (**c**) shows the same sample viewed under UV fluorescence lamp (analysis and photograph: Fernández M.O.). Detail. Private collection.

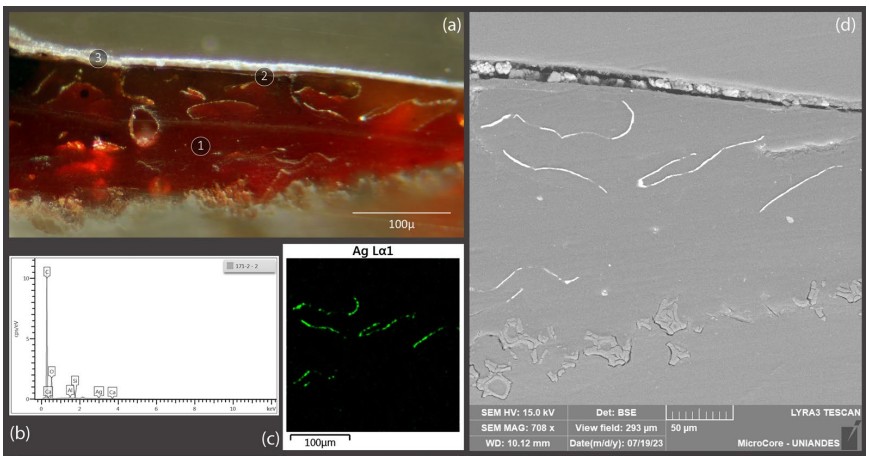

**Figure 12.** The cross-section (**a**) shows a 'venturine' layer (1). The SEM image (**d**) shows the thin pieces of silver within the resin, generated by kneading the resin after the addition of the silver leaf. Layer 2 is an orange-coloured resin, and layer 3 a silver leaf coloured composite membrane. The EDS spectrum (**b**) and EDS map (**c**) corroborate the presence of pure silver (Ag) (analysis and photograph: Fernández M.O.). Detail. Private collection.

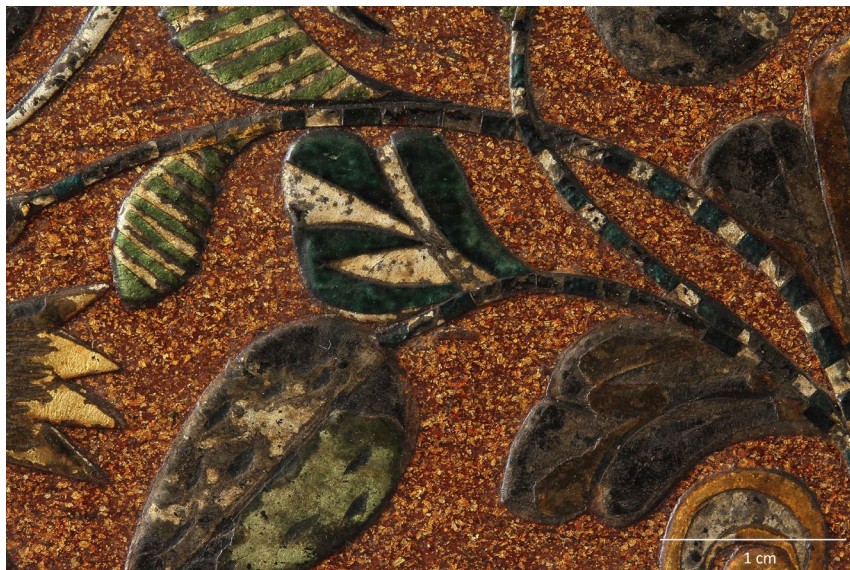

**Figure 13.** The effect created by the 'venturine' used as a background for the composition generates a sparkling finish that contrasts with the decorative details applied on top. The latter are made with silver leaf sandwiched between two resin membranes that, depending on the dye given to the upper one, acquire a specific hue. The metal leaf used in this way acquires a slightly different optical effect from the background. (Photograph: Álvarez-White, M.C). Detail. Private collection.

### 3.5. The Appearance of Other Metals

Some pieces from the Republican period [ca.1819–1886] have been dated by observing the presence of iconographic elements such as the different coats of arms that were used in what is now the Republic of Colombia. A significant change during this period was the introduction of brass, which was used alongside silver leaf.

Brass consists of a copper and zinc alloy that is used in thin metal sheets coated with translucent layers of resin (Figure 14). Brass is applied through an encapsulation process similar to that described for silver complex membranes, and provides a different iridescent metallic hue, extending the range of colour.

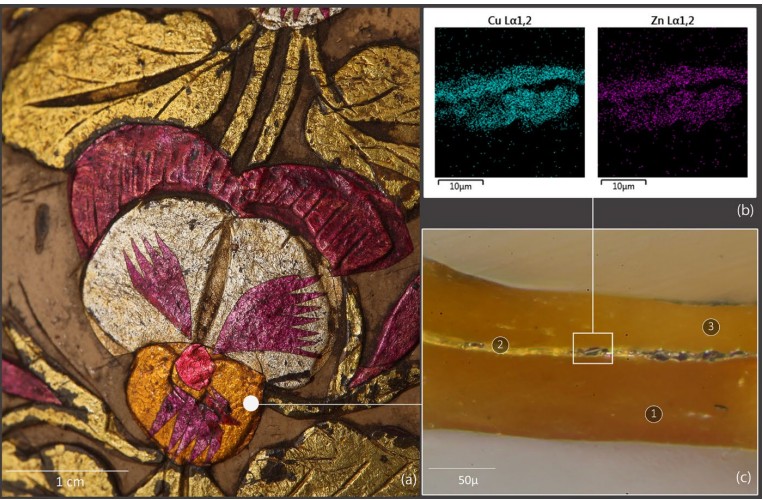

**Figure 14.** The lower petal, with a coppery finish, is achieved with brass leaf (**a**), while the iridescent whites and the golden highlights on the leaves were made with thin silver leaf. The white dot in this image indicates the sampling location (Photograph: Álvarez-White, M.C.). As can be seen in the elemental mapping (**c**) of a stratigraphic section (**b**) at 400×, the metallic layer (2) is an alloy of copper and zinc (Cu-Zn). Layers 1 and 3 correspond to the coloured resin layers on figure (**c**) (Analysis and photograph: Fernández, M.O).

Pieces from this period often have light opaque backgrounds that use lead white. Iridescent decorations in various shades are applied over the white background to achieve, among other things, the coppery effects typical of brass and the golden effect obtained with the yellow-tinted *mopa-mopa* covering silver.

## 4. Conclusions

Difficulties in dating pieces decorated with *barniz de Pasto* create obstacles to establishing a precise chronology. Many of the pieces have lost basic information such as their provenance, many are scattered in private collections, and even the few pieces that can be found in museums are of unclear origin.

By the time the Spaniards arrived, the indigenous populations were already familiar with resin and were skilled in using it. Interest in *mopa-mopa* objects led to the need for production to meet demand. The objects were made to order for the political and religious elite, who sometimes sent them abroad.

The calcium carbonate used in beads in the pre-Hispanic period was replaced by lead white, a material imported from Spain which the *barnizadores* adapted for use in the technique by mixing it, in powder form, with resin. This white pigment is common in early pieces, possibly made at the beginning of the 17th century, when there was already a settlement of people who could dedicate themselves to this work. The continuity of the use of lead white is observed from the arrival of the Spaniards until the Republican period.

One could postulate that the use of silver leaf was introduced towards the second half of the 17th century and that it constituted an important aspect in the evolution of the technique, as it was adapted with unique characteristics. The thin metallic leaf was used in the form of composite membranes. Using heat and manual pressure, it was applied directly to the surfaces of the new objects without any preparatory base. At this stage, the range of colour was also increased by using organic dyes, whose transparency, once mixed with the resin, made it possible to obtain metallic glitter in different shades in addition to the iridescent metallic tone of the sheet.

The calomel used in the *barniz de Pasto* gives the decoration a white finish that is different from the opaque one provided by lead white. Interestingly, this material has been found in Europe, also exceptionally, in manuscripts and miniatures from the 15th and 16th centuries according to Crippa et al. [30] (p. 15).

The descriptions from the chroniclers and the research carried out so far both identify the brilliance as a characteristic and recurrent element in the 18th century, which is when the technique reached its greatest splendour. During this period, the use of ground silver leaf known as 'venturine' introduced a novel effect in the *barniz de Pasto* to create a particular sparkling finish. This method is an innovative way of using silver leaf that has only been found in pieces decorated with this technique, so far.

In the Republican period, which is just beginning to be systematically studied, the presence of brass leaf has been documented. It is applied in a similar fashion to the encapsulation of silver into complex membranes but gives a different shade of coppery finish as seen in image (a) in Figure 14. Objects decorated with brass leaf often have a resin background pigmented with lead white.

There are many aspects of the *barniz de Pasto* technique that have yet to be investigated, including: when exactly silver leaf and calomel started being used and when they ceased to be used, identifying the dyes in the laboratory, foreign influences on the decoration, and the impacts on the health of the craftsmen due to the direct ingestion of lead and mercury from chewing them to amalgamate them with resin.

**Supplementary Materials:** The following supporting information can be downloaded at https://www.mdpi.com/article/10.3390/heritage6100344/s1. Supplementary Materials S1: original text in Spanish.

**Author Contributions:** Conceptualisation: M.C.Á.-W. Writing, preparation of the draft and graphic design of the figures: D.C. Analysis: M.O.F. Writing, revision, and editing: M.C.Á.-W. All authors have read and agreed to the published version of the manuscript.

**Funding:** This research received no external funding.

**Data Availability Statement:** Not applicable.

**Acknowledgments:** We kindly thank the Universidad de los Andes and the Support Fund for Assistant Professors for the provision of internal funding for the multispectral imaging and MEB-EDS analyses. The authors also thank Chiara Perera Jayasekarage and Izzy Seah from University College London (UCL) for their translation into English of the Spanish original version of this article.

**Conflicts of Interest:** The authors declare no conflict of interest.

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
