# Peer review of "The Splendour of Glitter: Silver Leaf in barniz de Pasto Objects"

_heritage, doi:10.3390/heritage6100344_

Round 1
Reviewer 1 Report
This paper makes a major contribution not just due to its illustration of the different ways metal leaf was used in Barniz de pasto, but also for the most thorough and detailed references to early chroniclers' mention of techniques and materials that I have seen. Since the authors elect to reference 16th century and early 17th century sources, I believe that the bibliography on Andean keros could be relevant to consult where such publications illuminate contemporary evidence of colorants used with mopa mopa and therefore highly likely for Barniz de pasto. I include two of these references below:
this one for its confirmation of cochineal wit mopa mopa and a discussion of its appearance over time:
Pearlstein, E., Mark Mac Kenzie, Emily Kaplan, Ellen Howe, Judith Levinson 2015, Tradition and Innovation, Cochineal and Andean Keros, in A Red Like No Other: How Cochineal Colored the World, edited by Carmella Padilla and Barbara Anderson, Rizzoli, NY and Museum of International Folk Art, Santa Fe, New Mexico, 44-51
and this one for its discussion of lead white pigments and their source on Andean keros:
Allison N. Curley, Alyson M. Thibodeau, Emily C. Kaplan, Ellen Howe, Ellen Pearlstein, Judith Levinson, 2020, Isotopic composition of lead white pigments on qeros: Implications for the chronology and production of Andean ritual drinking vessels during the colonial era, Heritage Science 8:72, https://heritagesciencejournal.springeropen.com/articles/10.1186/s40494-020- 00408-w
These references add some dimensions about the materials being described:
Katz, M. Colonial Spanish American Lacquered Objects in the Collection at the Hispanic Society of America. In Wooden Artifacts Group Postprints, Presentations from the 2016 AIC Annual Meeting in Montréal, Canada; AIC: Montreal, QC, Canada; pp. 37–48. Available online: https://www.culturalheritage.org/docs/default-source/publications/periodicals/wooden-artifacts-group/wooden-artifacts-group-postprints-vol-30-2016.pdf?sfvrsn=2c6b0a20_8
Newman/Derrick 2002: R. Newman/M. Derrick, Painted Qero Cups from the Inka and Colonial Periods in Peru. An Analytical Study of Pigments and Media. Materials Research Society Proceedings 712 (Warrendale 2002).
I congratulate the authors on a richly illustrated and articulated article.
Certain terminology, such as imprisoning metal leaf between layers of Barniz de pasto, would benefit from revision. The authors could instead simply refer to the metal leaf as sandwiched or layered, as it is an intentional effect.
There are places in the references where full translation to English is not achieved, i.e. Newman y Derrick...
Author Response
Dear Reviewer 1. Thanks a lot for your insights and generous contribution to improving the manuscript. We respond point-by-point to your comments:
- We include the two references (Pearlstein et al. and Allison et al.) about using cochineal in mopa-mopa as we find them very useful.
- We include Katz et al. reference. Great suggestion.
- We do not include Newman/Derrik's paper because we couldn't find a complete copy of this reference to check their contents and see how to directly link it to the manuscript.
- We do revise the terminology and used sandwiched instead of the long initial sentence. Completely agree with this comment.
- We change the references not completely translated. Thanks a lot.
We submit a new version. We highlight the changes in red for you to easily detect them. You may find the second reviewer's suggestions included as well.

Reviewer 2 Report
It would be nice to add some raking light photos of the objects, also as macro, in order to appreciate the surface relief of this technique.
Also, it would be nice to have some images of the actual items, so, as for me that have no experience with this technique, I can see how they look like. These items are mentioned in the paper, but not shown in their entirety, unless as just tiny details.
I noticed the manuscript is missing the usual “Conclusion ” section. This is a mandatory part of a paper. I think the current “Discussion ” section could be renamed “Conclusions” because the section is indeed a summary of the findings.
Author Response
Dear Reviewer 2. Thanks a lot for your useful comments. We completely agree with them all. We respond point-by-point as follows:
- We include a new macro photograph (Figure 7) with raking light taking your advice to better show the relief of the surface.
- We include a full object image (Figure 6) of a small desk from the 18th century (ca). Great suggestion.
- We changed the "Discussion" part to "Conclusion".
We send you the new version of the manuscript with the changes highlighted in red. You may find the first reviewers comments as well.
